# An Operational Atmospheric Correction Framework for Multi-Source Medium-High-Resolution Remote Sensing Data of China

**Hao Zhang** [1], **Dongchuan Yan** [2], **Bing Zhang** [1,3,*], **Zhengwen Fu** [1], **Baipeng Li** [1] and **Shuning Zhang** [1]

1   Aerospace Information Research Institute, Chinese Academy of Sciences, No. 9 Dengzhuang South Road, Beijing 100094, China
2   Institute of Mineral Resources Research, China Metallurgical Geology Bureau, Beijing 101300, China
3   University of Chinese Academy of Sciences, No. 19(A) Yuquan Road, Shijingshan District, Beijing 100049, China
*   Correspondence: zhangbing@aircas.ac.cn; Tel.: +86-10-8217-8002; Fax: +86-10-8217-8177

**Abstract:** Land surface reflectance (LSR) data form the basis of quantitatively remotely sensed applications. For accurate LSR retrieval, atmospheric correction has been investigated by many researchers and implemented in typical processing systems, including common atmospheric correction software for various types of datasets and automatic operating systems for application to certain individual data sources. In recent years, China has launched multiple medium–high-resolution satellites but has not provided standard LSR products partly because of the lack of an appropriate operational system. In this paper, a multi-source remote sensing LSR product system for medium- and high-resolution data is proposed, called the "Operational Atmospheric Correction Framework for multi-source Medium-high-resolution Remote Sensing data of China" (ACFrC). The AC algorithm, processing flow, and design of the multi-source LSR system were described in detail. A practical atmospheric correction algorithm was proposed specially for data in only the visible and near-infrared (VNIR) bands. The entire processing chain was divided into modules for multi-source data ingestion, apparent reflectance calculation, cloud and water identification, atmospheric correction, and standard LSR product generation. To date, most types of multi-source data have been tested using the ACFrC system, with reasonable results being obtained. From the preliminary results, the 313 scenes of LSR products from the GaoFen-2 (GF-2) satellite over China for the period from 2015 to 2018 were cross-compared with Landsat-8 LSR acquired on the same day, showing an overall uncertainty less than 0.112 × LSR + 0.0112. Further, the ACFrC data processing efficiency was found to be suitable for automatic operation. System improvement is ongoing and future refinements will include online cloud parallel computing functionality and services, more robust algorithms, and other radiometric processing functions.

**Keywords:** atmospheric correction; land surface reflectance products; look-up tables (LUTs); medium–high-resolution data

## 1. Introduction

A great number of medium–high-resolution satellites have been launched in recent years and related mission plans are continuously being developed. This development is driving the scientific community to implement regional or global quantitative remote sensing applications using various medium–high-resolution satellites. Since 1972, the Landsat program has supplied the world's longest continuous series of medium-resolution satellite data, provided by Landsats 1 to 8 [1]. The Landsat 9 satellite was launched on 27 September 2021 and joins Landsat 8 in orbit; this has reduced the repeat coverage to 8 days [2]. The European Space Agency (ESA) launched two medium-resolution satellites in 2015 and 2017, called "Sentinel-2A" and "-2B", respectively. These two Sentinel satellites, which provide

continuity for Satellite Pour l'Observation de la Terre (SPOT)- and Landsat-type image data, have enhanced spectral band performance, wider swaths (~290 km), and shorter revisit periods (~5 d) than older satellites. In a previous study, the medium-resolution temporal observation frequency was found to be much improved by combining Landsat-8 with Sentinel-2A and -2B, with a global median average revisit interval of 2.9 days [3]. Meanwhile, an increasing number of commercial high-resolution satellites (Worldview-2 and -3, Geoeye, Rapid-eye, etc.) routinely provide data for various applications in fields such as water resources [4], wetland [5], agriculture [6,7], and forests [8]. To shorten the revisit period, similar satellites are operated as a single constellation, such as the Digital-Globe [9], Airbus [10], SkySat [11], PlanetScope [12], and Jilin-1 constellations [13]. In the future, hundreds or thousands of high-resolution satellites will greatly enhance the speed of earth surface imaging. Therefore, a number of researchers and agencies have initiated global or regional applications based on medium–high-resolution satellite data, such as applications for land cover classification [14], land cover change monitoring [15,16], water resource monitoring [4], crop mapping [17], global change monitoring [18,19], and regional change monitoring [20].

The surface reflectance (SR) products derived from satellite remote sensing data constitute the basis of many applications. The key step during the SR retrieval process, called atmospheric correction (AC), aims to minimize the absorption and scattering influence of aerosol and atmospheric molecules. A number of AC-related algorithms have been developed in past decades, such as the dark dense vegetation (DDV) method or ark target (DT) method and its refinements [21–23], the deep blue (DB) method [24,25], the multi-angle implementation of atmospheric correction (MAIAC) method [26], and the multi-temporal (MT) method [27]. The DDV method has been widely used and has been implemented in AC software packages, including Atmospheric and Topographic Correction (ATCOR) [28,29], Atmospheric Correction Now (ACORN) [30], Fast Line-of-sight Atmospheric Analysis of Hypercubes (FLAASH) [31–33], and High-accuracy Atmospheric Correction for Hyperspectral Data (HATCH) [34]. MODIS SR products are retrieved using the improved DDV algorithm and lookup tables constructed by the second simulation of a satellite signal in the solar spectrum vector (6SV) code [35], with aerosol optical thickness (AOT) retrieval also being achieved through use of the more robust empirical relationship between the 470- and 670-nm bands. The aerosol type is estimated from four pre-assigned types using four other bands (i.e., 490, 443, 412, and 2130 nm) [22]. The Climate Change Initiative sponsored by the ESA since 2010 provides global SR products derived from the Medium-Resolution Imaging Spectrometer (MERIS) for the period of 2003 to 2012 [36]. MERIS SR products are derived using an artificial neural network method, based on the matrix-operator model (MOMO) radiative transfer (RT) code [37]. The Flemish Institute for Technological Research in Belgium has also produced SPOT vegetation SR products, with the latest version of Collection 3 being available at present [38].

As more and more access to medium–high-spatial-resolution satellite data becomes available, SR products derived from such data and their associated product systems become hot topics. Previous studies have illustrated the AC methods aiming at Landsat Thematic Mapper (TM)/Enhanced Thematic Mapper Plus (ETM+) by look-up tables (LUTs) methods based on LOWTRAN 6, LOWTRAN 7, and MODTRAN codes [39–42]. Some researchers have developed practical AC methods for large-scale medium-resolution data restricted to certain regions. Targeting the radiometric processing of large-scale Landsat data, a practical scheme to realize AC after AOT retrieval over persistent dark objects (i.e., water bodies) and topography correction, has been developed through the application of a modified C-correction [43]. Some investigators have presumed the aerosol properties at certain locations. For instance, an operational SR system for Landsat TM/ETM+ and SPOT High-resolution Geometric (HRG) in Eastern Australia has been developed, which considers the atmospheric, topographic, and bidirectional effects throughout the entire processing chain [44]. In the relevant study, the AOT at 550 nm was set to 0.05 and continental aerosol was assumed. However, although this method is suitable for application

in certain regions, the aerosol characteristics must be estimated for application in other areas. Currently, the SR products for Landsat data routinely provided by the U.S. Geological Survey are derived based on two core algorithms. One is specially tailored towards Landsat 4/5/7 data and implemented in the Landsat Ecosystem Disturbance Adaptive Processing System (LEDAPS) [45]. This algorithm can be classed as a typical DDV algorithm based on the empirical linear relationship between the red and blue band, assuming a fixed "continental" aerosol model. The algorithm uses auxiliary data, including atmospheric sea level pressure and water vapor data from the National Centers for Environmental Prediction (NCEP) and the National Center for Atmospheric Research (NCAR), ozone data from the National Aeronautics and Space Administration (NASA) Earth Probe Total Ozone Mapping Spectrometer (EP TOMS), and the 0.05° digital elevation model (DEM). The other algorithm was developed especially for application to Landsat 8 and Landsat 9 Operational Land Imager (OLI) data, assuming a fixed "clear urban" aerosol type [46]. It derives the AOT based on the relationship between the vegetation index and blue and deep blue bands and requires the same auxiliary data as the previous algorithm. In addition, the NASA Web-enabled Landsat Data (WELD) project derives SR products from Landsat 7 using MODIS-derived atmospheric parameters, because their overpass time differs by approximately half an hour [47]. The WELD project has been extended to provide Landsat SR globally for six three-year epochs spaced at five-year intervals from 1985 to 2010 [48]. Apart from the Landsat SR products, commercial satellite corporations (e.g., Planet) have begun to provide high-resolution SR products. The Planet satellite data are atmospherically corrected based on vector 6S LUTs, using the same atmospheric parameters (from MODIS Aerosol Optical Thickness (MOD09CMA) products) for each scene and assuming the "continental" aerosol model, while neglecting the adjacency, haze, and thin cirrus effects [49]. Similarly, the MODIS and VIIRS atmospheric products were used as inputs to derive the CBERS-4 SR products at a continental scale with the acceptable accuracy [43]. To combine the merits of DDV method and MT method, a prospective processor called MACCS-ATCOR Joint Algorithm (MAJA) was proposed to retrieve SR from Sentinel-2 by using the Copernicus atmosphere aerosol forecasts to set the aerosol type to improve the retrieval accuracy [50]. There are have also been various academic AC processors developed by different groups, such as ACOLITE, CorA, FORCE, LAC, etc., which have been used in the AC inter-comparison exercise for dealing with Landsat 8 and Sentinel-2 datasets [51].

China has also launched many medium- and high-resolution satellites in recent years, as part of the China–Brazil Earth Resources Satellite (CBERS), Huan Jing (HJ), Ziyuan (ZY), Beijing (BJ), and Gaofen (GF) series, with numerous future missions being planned. However, standard SR products derived from these multi-source satellite data are not yet available; the difficulty in deriving standard SR products has greatly hindered the quantitative application of domestic satellite data in China.

In this study, an operational atmospheric system called the "Operational Atmospheric Correction Framework for multi-source Medium-high-resolution Remote Sensing data of China (ACFrC)" was proposed for application to these multi-source medium- and high-resolution satellite data of China, so as to produce land SR (LSR) products. Although the software was mainly designed for Chinese multi-source data, other international satellite data were also included, especially the data with only 4 bands. However, this software was not intended for the AC of hyperspectral remote sensing data, which have been discussed thoroughly in a literature review [52]. The algorithm rationale, integration, and implementation, as well as the design of the multi-source SR system are described in detail.

## 2. Datasets

ACFrC was especially designed to deal with multi-spectral satellite data without ShortWave InRed (SWIR) bands, including those from HJ-CCD, ZY02C, ZY3, GF1, GF2, THEOS, CEBERS-01, -02, -02B, and -4. The detailed specifications pertaining to these data are listed in Table 1.

**Table 1.** Spatial and spectral specifications for medium- and high-resolution satellite data (NIR: near infrared).

| Satellite | Spatial Res. (m) | Spectral Bands (μm) | | | |
|---|---|---|---|---|---|
| | | **Blue** | **Green** | **Red** | **NIR** |
| THEOS | 15 | 0.45–0.53 | 0.53–0.60 | 0.62–0.69 | 0.77–0.69 |
| SPOT6 | 6.0 | 0.455–0.525 | 0.53–0.59 | 0.625–0.695 | 0.76–0.89 |
| GeoEye-1 | 1.64 | 0.45–0.51 | 0.51–0.58 | 0.655–0.690 | 0.78–0.92 |
| Advanced Land Observing Satellite (ALOS) | 10 | 0.42–0.50 | 0.52–0.60 | 0.61–0.69 | 0.76–0.89 |
| HJ-1 (A & B) | 30 | 0.43–0.52 | 0.52–0.60 | 0.63–0.69 | 0.76–0.90 |
| CBERS-01/02 | 19.5 | 0.45–0.52 | 0.52–0.59 | 0.63–0.69 | 0.77–0.89 |
| CBERS-02B | 20 | 0.45–0.52 | 0.52–0.59 | 0.63–0.69 | 0.77–0.89 |
| CBERS-04 [1] | 10/20 | 0.45–0.52 | 0.52–0.59 | 0.63–0.69 | 0.77–0.89 |
| GF-1 [2] | 8/16 | 0.45–0.52 | 0.52–0.59 | 0.63–0.69 | 0.77–0.89 |
| GF-2 | 4 | 0.45–0.52 | 0.52–0.59 | 0.63–0.69 | 0.77–0.89 |
| ZY-3 | 6 | 0.45–0.52 | 0.52–0.59 | 0.63–0.69 | 0.77–0.89 |
| ZY-3-02 | 5.8 | 0.45–0.52 | 0.52–0.59 | 0.63–0.69 | 0.77–0.89 |

[1] Spatial resolutions of 10 m for green, red, and NIR bands, and of 20 m for four VNIR bands; [2] two sensors with the same bands have spatial resolutions of 8 and 16 m, and swaths of 60 and 800 km, respectively.

The above satellites typically have 4 bands, and their SR products are difficult to access, like Sentinel- or Landsat-based ones. Therefore, an operational AC algorithm was proposed specially to deal with the 4 bands of data given above, so as to meet the operational running requirements for ACFrC.

## 3. Algorithm Strategy

The most important step in AC imagery is to determine the water vapor distribution and the aerosol property for the remotely sensed data from the visible to shortwave wavelength range under study. Water vapor is commonly represented by the columnar water vapor (CWV) abundance derived from the water vapor absorption bands and adjacent non-absorption bands through methods such as the continuum interpolated band ratio (CIBR) [53,54], 3-channel ratio [55], and atmospheric pre-corrected different absorption (APDA) [56] techniques. Some researchers have also reported that the results from these different water vapor retrieval methods are comparable [57]. However, the input data for the current version of ACFrC did not have water vapor absorption bands and the CWV abundance was not derived from the data themselves. Instead, the CWV (along with the sea-level atmospheric pressure) data were obtained from NCEP and NCAR. Two aspects of the aerosol property are needed for AC: the aerosol type (or model) and the AOT. The aerosol type is a pre-requisite parameter before AOT retrieval and is difficult to determine from multi-spectral data themselves. Generally, this parameter is always assumed based on either observations at numerous in situ sites or researcher experience. A universally acceptable aerosol type assignment method is based on observations from the Aerosol Robotic Network (AERONET), which vary by season and location [58]. However, in other studies, a fixed global aerosol type is often arbitrarily assumed, such as the "continental" aerosol type, internally defined in 6SV by LEDAPS (Masek et al., 2006), and the "urban clean" dynamic aerosol type used in operational OLI SR retrieval [46]. The OLI SR operational algorithm, short for LaSRC, use the ratio between the red and blue bands from the Landsat 8–9 ratio map file to minimize the AOT inversion residual by three bands (i.e., the two blue bands and the red band) [46]. The algorithm principle was extended to 4-band optical data by revising the original procedure to two aspects, one being the vegetation index, replaced by the common NDVI quantity (Equation (1)), and the other being the inversion residual, computed by reflectance of the red and blue band (Equation (2)).

$$NDVI = \frac{\rho_{nir} - \rho_{red}}{\rho_{nir} + \rho_{red}} \tag{1}$$

$$\Delta = |\rho_{blue} - r_{blue}\rho_{red}| \tag{2}$$

where $\rho_{blue}$, $\rho_{red}$, and $\rho_{nir}$, respectively, represent the SR of the blue (0.49 µm), red (0.66 µm), and near infrared (0.83 µm) bands, and $r_{blue}$ is the ratio between the blue band and the red band.

The aerosol type selection, AOT retrieval, and AC are detailed in the following subsections.

### 3.1. Aerosol Type Selection

Similar to the extensible Bremen aerosol retrieval (XBAER) algorithm [23], aerosol types based on a priori assumptions were prescribed globally for a given region and season from an analysis of AERONET observations. Three fine-dominated types and a coarse-dominated dust type were used in ACFrC, with a $1° \times 1°$ global grid, namely weakly, moderately, and strongly absorbing aerosol types and a dust type (Figure 1). The dust type was determined by the MODIS Land Cover product of 2019. Each of these four types were expressed by a bimodal lognormal function as a combination of fine and coarse models, with detailed micro-physical properties based on relevant work [58]. These properties of the assumed "dynamic aerosol" types were related to aerosol loading, with an AOT function implemented at 0.55 µm.

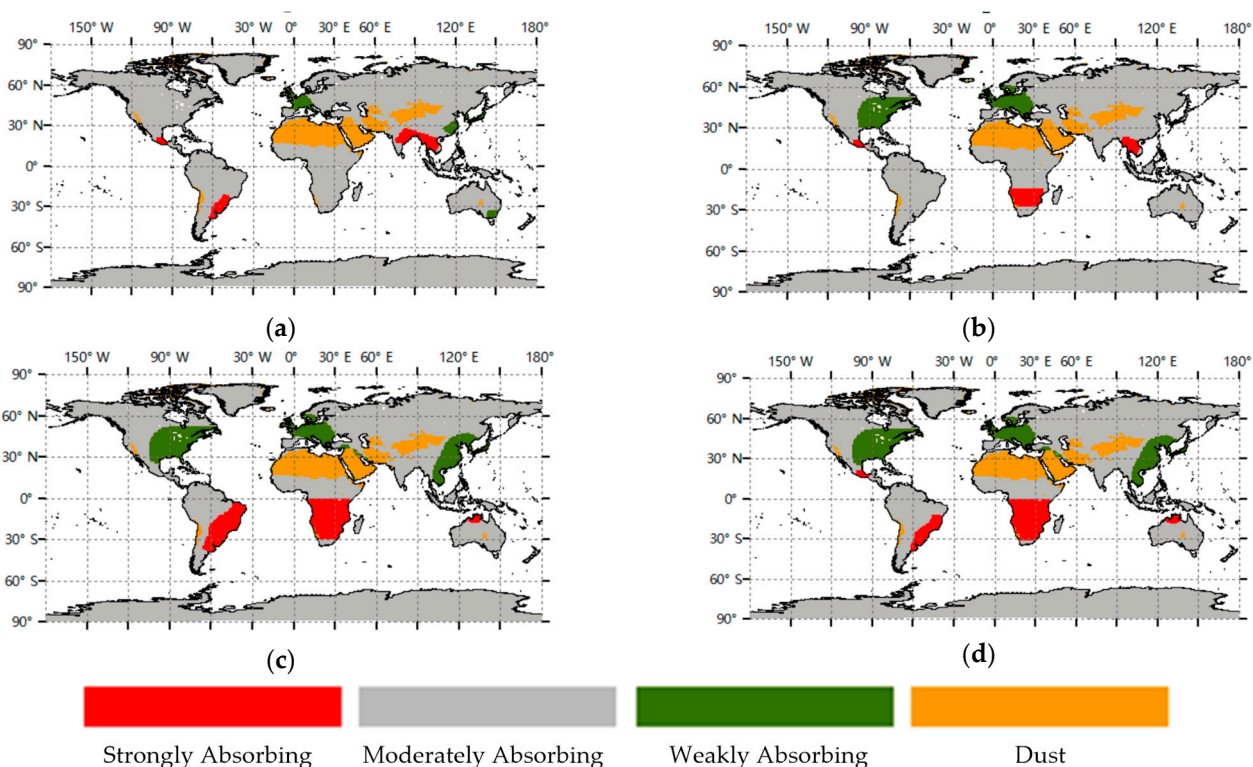

**Figure 1.** Aerosol types over land used in the ACFrC designated at $1° \times 1°$ grid for different seasons. The four sub-figures show four seasons: (**a**) December, January, and February (DJF); (**b**) March, April, and May (MAM); (**c**) June, July, and August (JJA); and (**d**) September, October, and November (SON).

### 3.2. AOT Retrieval

Figure 2 shows the scheme of the AOT inversion for the 4-band optical data. The key step was to determine the ratio between the red and blue bands for each pixel from the Landsat 8-9 ratio map file, which was derived from MODIS at 0.05°. To avoid the oversampling issue, the high-resolution TOA data were firstly aggregated into the medium scale (i.e., 20 to 30 m) and the band ratio for each aggregated pixel based on the linear function of NDVI was computed, expressed as

$$Ratio = a \cdot NDVI + b \tag{3}$$

where *a* and *b* are the coefficients from the Landsat 8-9 ratio map file, depending on the spectral band, and the *NDVI* is calculated from the surface reflectance in the red and NIR bands, assuming 0.05 of AOT at 550 nm.

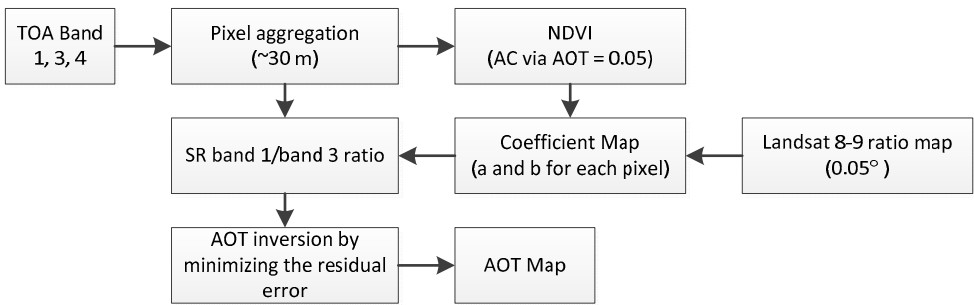

**Figure 2.** AOT inversion scheme for the optical data with 4 bands.

### 3.3. Atmospheric Correction

Besides the aerosol type and AOT, other auxiliary data (atmospheric pressure, water vapor, zone, and DEM) are also needed as inputs for AC. In the proposed system, the sea-level atmospheric pressure and water vapor, defined every 6 h at 2.5° spatial resolution, were obtained from NCEP and NCAR. The ozone data were acquired from two datasets: the NASA EP TOMS and the NASA National Oceanic and Atmospheric Administration (NOAA) Total Operational Vertical Sounder (TOVS) ozone retrievals. The former dataset was mainly used for the period before 2005 and the latter was taken as a substitute if EP TOMS data were unavailable. The TOVS data were also used for the period after 2005. The 1 km resolution Advanced Spaceborne Thermal Emission and Reflection Radiometer (ASTER) DEM was employed to adjust the atmospheric pressure from sea to surface level via a negative exponent function [35]. To standardize the process, all data types were first transformed into apparent reflectance images and corrected according to the gas absorption behaviors (for ozone, water vapor, and other gases), before being entered into the AC flow.

### 3.3.1. Imaging Geometry and Apparent Reflectance Computation

Imaging geometry involves the solar zenith (SZ, $\theta_s$), solar azimuth (SA, $\phi_s$), viewing zenith (VZ, $\theta_v$), and viewing azimuth (VA, $\phi_v$) angles. These angles are necessary for calculation of the apparent reflectance and AC but are not completely included in the dataset themselves. Thus, if they were not provided, they were determined according to the following strategy.

The SZ and SA angles were determined for each pixel via two steps: (1) these two solar angles were computed at a 1 km sampling distance over a spatial extent confined by four corner coordinates; (2) the solar angles for each pixel were assigned from the 1 km solar angle images via a bilinear resampling method. The state-of-the-art solar position algorithm (SPA) proposed by the National Renewable Energy Laboratory (NREL) was used here [59]. The VZ and VA were not considered as fixed angles, as in the case of images with narrow swaths (e.g., Landsat satellite images). Here, the VZ and VA angles were estimated roughly for each pixel according to the satellite orbit inclination ($\varsigma$), off-nadir angle ($\eta$), and altitude (*H*) (Figure 3).

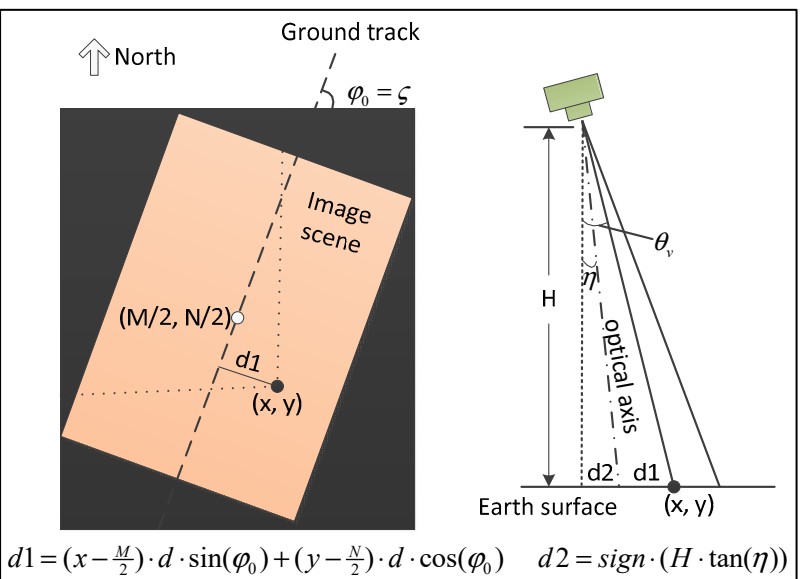

$$d1 = (x - \tfrac{M}{2}) \cdot d \cdot \sin(\varphi_0) + (y - \tfrac{N}{2}) \cdot d \cdot \cos(\varphi_0) \qquad d2 = sign \cdot (H \cdot \tan(\eta))$$

**Figure 3.** Diagram of the rough estimation of VZ of certain pixel position.

$$\tan(\theta_v(x,y)) = \frac{(x - \frac{M}{2}) \cdot d \cdot \sin(\varphi_0) + (y - \frac{N}{2}) \cdot d \cdot \cos(\varphi_0) + sign \cdot (H \cdot \tan(\eta))}{H} \quad (4)$$

where $x$ and $y$ denote the sample and row position of a certain pixel; $M$, $N$, and $d$ denote the sample and row numbers and ground sampling interval, respectively; and $\varphi_0$ is the angle measured from the north to the line parallel to the row direction, which is equal to $\varsigma$ or $180° - \varsigma$ in the case of a descending or ascending orbit, respectively. In addition, the sign is set to 1 or $-1$ for a descending or ascending orbit, respectively.

The viewing azimuth angle for each pixel is determined by the sign of Equation (4):

$$\phi_v(x,y) = \begin{cases} \varphi_0, & \tan(\theta_v(x,y)) > 0, \\ 0, & \tan(\theta_v(x,y)) = 0, \\ \varphi_0 + 180°, & \tan(\theta_v(x,y)) < 0 \end{cases}, \quad (5)$$

Most of the datasets were provided in the form of the raw digital numbers (e.g., for GF-1 and -2, and HJ-1). The apparent reflectance $\rho^*$ is computed from the following Equation [52]:

$$\rho^*(i) = \frac{\pi \cdot R^2 \cdot L(i)}{E_0(i) \cdot \cos(\theta_s)}, \quad (6)$$

where $E_0$ is the top of atmosphere (TOA) irradiance derived from a certain data source, especially referring to the irradiance with which the sensor was radiometrically calibrated; $i$ denotes a certain band; and $R$ is the Earth–Sun distance in astronomical units (AU).

### 3.3.2. Gaseous Transmission Correction

Similar to the MODIS AC process [35], $\rho^*$ is first corrected according to the gaseous transmission of the ozone ($T^i_{g_{O_3}}$), water vapor ($T^i_{g_{H_2O}}$), and other gases ($T^i_{g_{OG}}$), as follows:

$$\rho_c^*(i) = \frac{\rho^*(i)}{T^i_{g_{OG}}(m, P) \cdot T^i_{g_{O_3}}(m, U_{O_3}) \cdot T^i_{g_{H_2O}}(m, U_{H_2O})}, \quad (7)$$

where $m$ is the air mass computed as $1/\cos(\theta_s) + 1/\cos(\theta_v)$; $P$ is the atmospheric pressure; $U_{O_3}$ is the integrated ozone content; and $U_{H_2O}$ is the integrated water vapor content. Expressions for and details of the three kinds of gaseous transmissions are given by Vermote et al. [35].

### 3.3.3. Land Surface Reflectance (LSR) Inversion

Under the Lambertian uniform target assumption, the relationship between the LSR ($\rho_t$) and apparent reflectance after gaseous transmission correction is rewritten as [35]:

$$\rho_c^*(\theta_s, \theta_v, \varphi, P, Aer, U_{H_2O}, U_{O_3}) = \frac{\rho_{atm}(\theta_s, \theta_v, \varphi, P, Aer, U_{H_2O})}{T_{g_{H_2O}}(m, U_{H_2O})} + Tr_{atm}(\theta_s, \theta_v, P, Aer)\frac{\rho_t}{1 - S_{atm}(P, Aer) \cdot \rho_t} \quad , \tag{8}$$

where $\rho_{atm}$ is the atmospheric intrinsic reflectance; $Tr_{atm}$ is the total atmosphere transmission (downward and upward); $S_{atm}$ is the atmospheric spherical albedo; $\varphi$ is the relative azimuth (AZ) angle (i.e., the difference between the solar and viewing azimuth angles); and *Aer* denotes a certain aerosol type determined by the AOT, aerosol single scattering albedo, and aerosol phase function.

The parameters in Equation (8) are computed from the following six quantities [35]:

(1)     $\tau_R(P_0)$: The Rayleigh optical thickness at standard pressure.

This quantity, computed using an approach with a relatively high accuracy, varies slightly with geographic location [60]. Accordingly, the Rayleigh optical thickness at actual pressure $\tau_R(P)$ is simply calculated by adjusting the pressure:

$$\tau_R(P) = P/P_0 \cdot \tau_R(P_0). \tag{9}$$

The Rayleigh optical thickness is used as an input parameter to compute the molecular or Rayleigh atmospheric reflectance $\rho_R(\theta_s, \theta_v, \varphi, P)$ and the Rayleigh spherical albedo using the CHAND.f and CSALBR.f subroutines of 6SV code, respectively.

(2)     $\rho_{R+Aer}(\theta_s, \theta_v, \varphi, P_0, Aer)$: The intrinsic reflectance at standard pressure.

The values of this quantity are pre-computed in 6SV for each band and each aerosol model under different SZ, VZ, and AZ angles.

(3)     $T_{atm}(\theta_s, P_0, Aer)$: The atmospheric transmission at standard pressure.

The values of this quantity are computed for each band and each aerosol model under different solar or VZ angles.

(4)     $T_{atm}(\theta, P_0, Aer)$: The atmospheric spherical albedo at standard pressure.

This quantity, which is dependent on the aerosol properties, is pre-computed by the 6SV code.

Thus, the values of three quantities, $\rho_{R+Aer}$, $\rho_R$, and $S_{atm}$, are pre-computed by the 6SV code and stored in LUTs. When applied to AC, these values should be interpolated by one-, two-, or even multi-dimensional tables for a certain input condition.

## 4. LUT Design

For feasible operation, the proposed system used LUTs to contain the atmospheric scattering components pre-computed by 6SV (Version 2.1) for different imaging geometries and atmospheric conditions for the data acquired by the various sensors listed in Table 1. Compared to the scalar mode of 6S, the 6SV enabled accounting for radiation polarization. It could eliminate the 5–10% errors caused by ignoring the polarization effects [61]. The scattering components, such as the Rayleigh optical depth, intrinsic atmospheric reflectance, upward and downward transmittance (diffuse plus direct), total spherical albedo, and Rayleigh spherical albedo, were stored in LUTs calculated for the following conditions: an SZ angle range from 0 to 72°; a VZ angle range from 0 to a maximum of 72°, determined according to the sensor's field of view (FOV); an AZ angle range from 0 to 180° with a 30° interval; and an AOT at 550 nm range from 0.01 to 2.0, with increments of 0.05 and 0.1 between 0.01–0.3 and 0.3–2.0, respectively. The values of VA and VZ were referred to in a previous work by Gao et al. [62]. The nodes of the five input parameters of the LUTs are listed in Table 2, and the other parameters were set to their fixed values in the standard

atmosphere. These included the solar azimuth angle ($0°$), the ozone content (300 DU), the water vapor content ($1.0 \, \text{g/cm}^2$), and the altitude (0 km). The coefficients of the empirical formulas of gaseous transmission were also pre-computed by fitting the transmission using factors such as the gas content, pressure, and imaging geometry.

**Table 2.** Dimensionality and LUT nodes.

| Factors | Node | Counts |
|---------|------|--------|
| Bands | Number of bands for special sensor | $x$ [1] |
| Aerosol type | Weakly, moderately, and strongly absorbing; dust | 4 |
| SZ angle (°) | 1.5, 12, 24, 36, 48, 54, 60, 66, 72 | 9 |
| AZ angle (°) | 0, 30, 60, 90, 120, 150, 180 | 7 |
| VZ angle (°) | Maximum nodes of 0, 12, 24, 36, 48, 54, 60, 66, 72 | $y$ [1] |
| AOT @ 550 nm | 0.01, 0.05, 0.10, 0.15, 0.20, 0.30, 0.40, 0.60, 0.80, 1.00, 1.20, 1.40, 1.60, 1.80, 2.0 | 15 |

[1] The values of $x$ and $y$ were determined by the sensor specifications. Taking HJ-1 as an example, $x$ equals 4 for its four bands (blue, green, red, and NIR) and $y$ equals 4 for its FOV of $52°$, corresponding to a maximum VZ angle ($0°$, $12°$, $24°$, and $36°$) just exceeding FOV/2.

For AC convenience, the LUTs generated by the 6SV code were specified for each remote sensor and stored in binary code with a seven-dimensional structure. The dimensional variables were the band index, aerosol type, VZ, AZ, and SZ angles, AOT at 550 nm, and output scattering components (i.e., $\rho_{R+Aer}$, $\rho_R$, and $S_{atm}$). When using the LUTs, a strategy of multiple dimensional linear interpolation was proposed according to the independent variables in the following order: the AOT at 550 nm, the AZ angle, the cosine of the VZ angle, and the cosine of the SZ angle.

## 5. System Design and Implementation

The entire processing chain is divided into modules for data reading, apparent reflectance, cloud and water masking, AC, and the LSR products, as shown in Figure 4. The tiling processing strategy is adopted to permit the input of extremely large data volumes. Necessary auxiliary data are also used, including radiometric calibration coefficients, atmospheric data (i.e., AOT, water vapor, and atmospheric pressure from MODIS products or NCEP reanalysis), and 1 km resolution ASTER DEM. The processing status is recorded once finished and a report is generated to trace any failure occurring in the entire processing chain. An order file in XML format is used to facilitate the processing of each image; this file contains all inputs, outputs, and parameter settings. The entire system is implemented in standard C++ language, which can maximize the platform portability, and part of the image processing module calls upon a universal remote sensing image processing library such as the Geospatial Data Abstraction Library (GDAL) [63].

Considering the differences in the metadata descriptions, image formats, and band sequences of the different data sources, data-dependent reading modules are developed for each individual dataset. Then, the apparent reflectance is computed with the assistance of auxiliary information including the imaging angles and radiometric calibration coefficients. The next step is to extract the cloud mask and water mask, which do not contain reliable AC values. Then, the AC module incorporating various AC algorithms can be used to generate the LSR. The final LSR in georeferenced tagged image file format (GeoTIFF) or hierarchical data format (HDF) is generated by the LSR product module. To facilitate automatic searching and downloading of the auxiliary data, the NCEP reanalysis and MODIS data are found according to the image acquisition time and the position of the pending image to assist in the AC process. For improved convenience, we pre-built a radiometric calibration coefficient database for the various remote sensors, with coefficients determined according to the sensor calibration time, name, etc. The calibration coefficients listed in this database can also be updated and maintained.

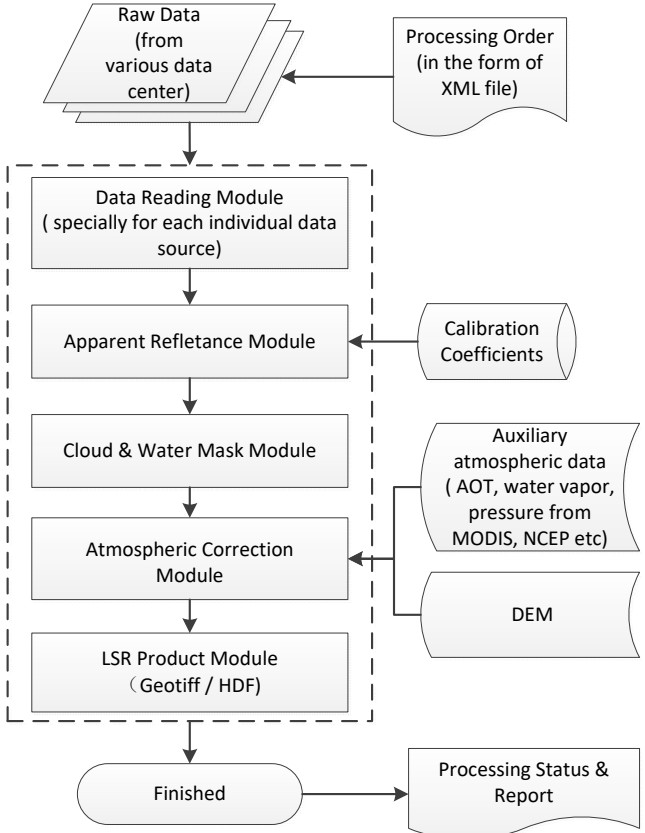

**Figure 4.** Data processing flow. The modules within the dashed rectangle are the core components of ACFrC.

In summary, the automatic production of LSR products was realized through the implementation of a matching strategy for the sensor and data. The final LSR products contain both the reflectance dataset and other auxiliary information, such as the masking layer and algorithm flag. Details of the masking layers and product format are given in the following subsections.

### 5.1. Cloud- and Water-Related Masking

As AC algorithms are mainly implemented over land surfaces, masking algorithms for cloud, water, shadow, and snow for application before AC have a long development history. Similar to the approach used in the ATCOR processing chain, we employed practical band index algorithms for most VNIR data [64]. The layer masking was implemented as follows.

Step 1. Background masking

AC is executed over all pixels except for the background pixels, which are always filled with a fixed value (e.g., 0 or $-9999$) outside the valid image region after geometric correction. The background pixel mask is applied if a pixel of any band has a value equaling the filled value.

Step 2. Water-related masking

The water masking strategy of ATCOR was adopted, which involves two cases for elevations lower or higher than 1.2 km [29]. If the surface elevation of a pixel is below 1.2 km, the water pixel criteria are set, i.e.,

$$\begin{aligned}\rho^*(blue) < 0.20 \text{ and } \rho^*(blue) > \rho^*(green) - 0.03 \text{ and} \\ \rho^*(NIR) < \rho^*(green) \text{ and } \rho^*(swir1) < T_{(water,swir1)}\end{aligned} \quad , \tag{10}$$

where $T_{water,swir1}$ is the water threshold (e.g., 0.03) in the SWIR1 band (approximately 1.6 μm), as defined by the users.

If the pixel elevation is higher than 1.2 km, the band criteria are set for the land surface temperature (LST), such that

$$\rho(NIR) < |T_{water,NIR}| \text{ and } \rho(SWIR1) < |T_{water,swir1}| \tag{11}$$

where $T_{water,NIR}$ is a predetermined parameter for the NIR band, similar to $T_{water,swir1}$ for the SWIR1 band. The SWIR1 criterion in Equations (10) and (11), i.e., $\rho^*(swir1) < T_{water,swir1}$, can be ignored if no SWIR1 band exists.

Step 3. Cloud-related masking

ATCOR uses band criteria to detect cloud and cloud shadows over water and land surfaces. For practical reasons, the cloud/shadow detection algorithm based on spectral indices (CSD-SI) was adopted in this study, as it provides good adaptation and robust performance for most multi-spectral sensors with both visible and infrared bands [64]. The core strategies include two spectral index calculations, i.e., for the cloud index (CI) and cloud shadow index (CSI), and the setting of corresponding threshold values. CI is computed based on the three visible bands (blue, green, and red), and the NIR band. The CSI is determined by the NIR band. Implementation of the CSD-SI algorithm is easy after the thresholds of the total eight parameters have been determined through fewer refining steps.

*5.2. LSR Product Format*

During the LSR retrieval, the TOA reflectance, AOT, cloud mask, and water mask are also derived and supplied as optional products for selection by the users. To standardize the file formats of the LSR products and the associated files derived from different types of remote sensing data, the LSR products are supplied in HDF and GeoTIFF formats, with file names composed of a combination of five string components as follows:

Part 1: A unique sensor name with varied string length is generated. For instance, "HJ1ACCD1" indicates the CCD1 sensor on the HJ1A satellite.

Part 2: The LSR spatial resolution in meters, represented by a varied string length, is given. An upscale resampling operation is conducted if the sensor has bands with different spatial resolutions, e.g., 20 m is assigned for the final LSR products retrieved from data from the CBERS-04 multi-spectral sensor. The unique sensor names and spatial resolutions for the different satellites and sensors from which the LSR products are retrieved are listed in Table 3.

Part 3: The scene acquisition time is given, represented by a string of 13 characters in length. This string includes the year, Julian day, hour, minute, and second in UTC time, i.e., "YYYYDDDHHMMSS".

Part 4: The path and row string is given, composed of six characters. The first three characters represent the path number and the remaining three represent the row number, i.e., "PPPRRR". These characters can be filled with zeros if the path and row numbers are unavailable.

Part 5: The flag, which indicates the physical meaning of the product, is given. For example, the extensions "toa", "lsr", "aot", "cld", "shw", "snw", and "wat" represent TOA reflectance, LSR, AOT, cloud mask, shadow mask, snow or ice mask, and water mask files, respectively, in GeoTIFF format. The "atc" extension is used for HDF format files in particular and contains the TOA reflectance at the minimum along with other optional output results.

**Table 3.** Unique sensor names and spatial resolutions for satellites and sensors from which LSR products are retrieved.

| Satellite | Sensor | Unique Name | Spatial Resolution (m) |
|---|---|---|---|
| THEOS | Multi-spectral (MS) | THEOS | 15 |
| SPOT6 | New AstroSat Optical Modular Instrument (NAOMI) | SPOT6 | 6.0 |
| GeoEye-1 | MS | GeoEye1 | 1.64 |
| ALOS | Advanced Visible and Near-infrared Radiometer Type 2 (AVNIR-2) | ALOS | 10 |
| HJ-1A | CCD1 | HJ-1ACCD1 | 30 |
| HJ-1A | CCD2 | HJ-1ACCD2 | 30 |
| HJ-1B | CCD1 | HJ-1BCCD1 | 30 |
| HJ-1B | CCD2 | HJ-1BCCD2 | 30 |
| CBERS-01/02 | CCD | CBERS | 19.5 |
| CBERS-02B | CCD | CBERS02B | 20 |
| CBERS-04 | Panchromatic and Multi-spectral (PMS) | CBERS04-PMS | 10 |
| CBERS-04 | Multispectral Camera (MUXCAM) | CBERS04-MUX | 20 |
| GF-1 | PMS | GF1-PMS | 8 |
| GF-1 | MUX | GF1-PMS | 16 |
| GF-2 | MUX | GF2-MUX | 4 |
| ZY-1-02C | PMS | ZY102C-PMS | 10 |
| ZY-3 | MUX | ZY3-MUX | 6 |
| ZY-3-02 | MUX | ZY3-02-MUX | 5.8 |

The five parts described above are joined together by dashed lines "_" and followed by the file extension (".tif" or ".hdf", depending on the file format). As an example, Table 4 lists the file names of the output products for one scene from the GF-1 satellite with the original ID "GF1_PMS1_E118.2_N38.9_20140623_L1A0000257917-MSS1."

**Table 4.** Examples of output file names for products retrieved from GF-1 scene.

| File Name | Specification |
|---|---|
| GF1-PMS_8_2014174032104_602090_atc.hdf | Contains attribution information and the data layers |
| GF1-PMS_8_2014174032104_602090_toa.tif | TOA reflectance |
| GF1-PMS_8_2014174032104_602090_lsr.tif | Land surface reflectance |
| GF1-PMS_8_2014174032104_602090_aot.tif | Retrieved AOT at 550 nm |
| GF1-PMS_8_2014174032104_602090_cld.tif | Cloud mask |
| GF1-PMS_8_2014174032104_602090_wat.tif | Water mask |

We used the HDF5 format to incorporate auxiliary information and data into one file. This file contains necessary attribute items such as the spatial resolution, production time, and several data groups, e.g., the imaging angle data, land surface reflectance, TOA reflectance, and mask layer groups. The structure and specifications of the HDF format file are described in Table 5.



**Table 5.** Specifications for output product in HDF5 format.

| Attribute | Specification | | |
|---|---|---|---|
| **Spatial Resolution** | **Spatial Resolution** | | |
| RawDataNames | Name list of all data files as inputs for target product retrieval | | |
| AcquisitionTime | Acquisition time in UTC for scene, expressed in "YYYYDDDHHMMSS" format | | |
| OrbitNum | Path and row numbers for scene, i.e., "PPPRRR" | | |
| StdProductName | Entire product name, consistent with file name | | |
| NumBand | Total number of bands in product | | |
| SpatialReference | Spatial reference represented by WKT string | | |
| DataGroupNum | Total number of data groups | | |
| Size | Spatial dimensions of sample and row, e.g., "15,412, 13,835" | | |
| ACAlgorithm | 4-band internal algorithm | | |
| Data group name | Data layer name | Data layer attribute | Specification |
| AngleData | ViewZenithAngle ViewAzimuthAngle SolarZenithAngle SolarAzimuthAngle | Scalefactor | Values derived by original value multiplying scale factor |
| | | IsImage | 0 or 1 indicating false or true |
| | | FillValue | Filled values for invalid pixels, e.g., "−32,768" |
| LandSurfaceReflectance | DataSet_1 DataSet_2 DataSet_3 ⋯ DataSet_x | BandID | Indicating band ID in original scene |
| | | SpectralRange | Spectral wavelength range in units of micrometer, e.g., "0.43, 0.52" |
| | | Scalefactor | Values derived by original value multiplying scale factor |
| | | FillValue | |
| TOAReflectance | DataSet_1 DataSet_2 DataSet_3 ⋯ DataSet_x | BandID | Indicating band ID in original scene |
| | | SpectralRange | Spectral wavelength range in units of micrometers, e.g., "0.43, 0.52" |
| | | Scalefactor | Values derived by original value multiplying scale factor |
| | | FillValue | |
| LayerMask | DataSet_CloudMask DataSet_ShadowMask DataSet_SnowMask DataSet_WaterMask DataSet_AOT | Scalefactor | Values derived by original value multiplying scale factor |
| | | FillValue | |

## 6. Preliminary Results

### 6.1. Product Accuracy

Regarding the product accuracy, most types of multi-source data have been tested by the system to date and reasonable results have been obtained. Two important performance indexes (i.e., the product accuracy and processing efficiency) related to the system were preliminarily evaluated based on a generation of many LSR products followed by evaluation. LSR products derived from GF-2 satellite data obtained over China (313 scenes in total) for the period of 2015 to 2018 were considered (Figure 5). The acquisition time of these data is listed in Figure 6, showing a minimum of six and maximum of sixty-two scenes. The selected 313 GF-2 scenes were representative of various types of surfaces, covering density forests, cities, crop lands, seashores, in-land waters, deserts, and arid lands over the mountainous and flat surfaces. Furthermore, the atmospheric conditions in these data

varied quite differently from very clear to thin clouds and even heavy clouds. The accuracy of these products was evaluated through cross-comparison with the Landsat-8 OLI LSR products, considered as the reference data set. The paired data of GF-2 and Landsat-8 should be acquired on the same day, assuming no variation in the surface. The differences between these two datasets needs to be treated appropriately in terms of spatial resolution, spectral bandpass, and sun-view geometry. Due to the small field of view of these two satellites and approximate acquisition time (no more than 1 h), the BRDF effects were thus ignored in the cross-compassion partly because of their similarities in sun-view geometry and partly because of the possible uncertainties when applying the BRDF compensation.

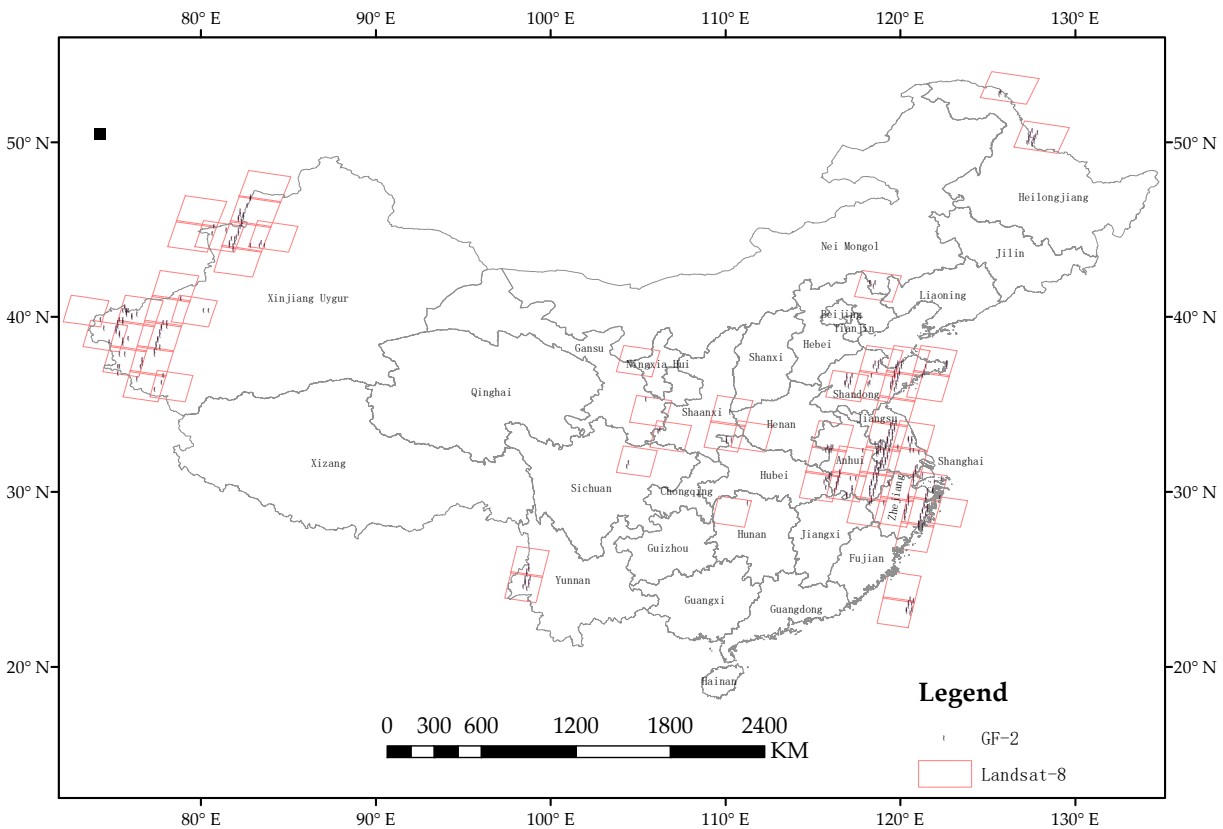

**Figure 5.** Map of the GF-2 scenes (purple dots) corresponding to the paired Landsat-8 scenes (red quadrangles) for cross-comparison over China and adjacent regions.

(1)　GF-2 3.2 m LSR spatial aggregation to 30 m resolution.

Firstly, the GF-2 pixels were spatially aggregated into Landsat 30 m spatial resolution to ensure the pixel-to-pixel comparison. Then, the following criteria were applied to select the valid GF-2 pixels: (i) cloud and cloud-shadow free pixels, (ii) snow- and water-free pixels, and (iii) saturation-free pixels. Landsat pixels comprising less than 100% valid GF-2 pixels were excluded from the analysis. The valid Landsat pixels were selected by the quality assessment files (i.e., * .QA_pixel.tif and * .QA_AEROSOL.tif).

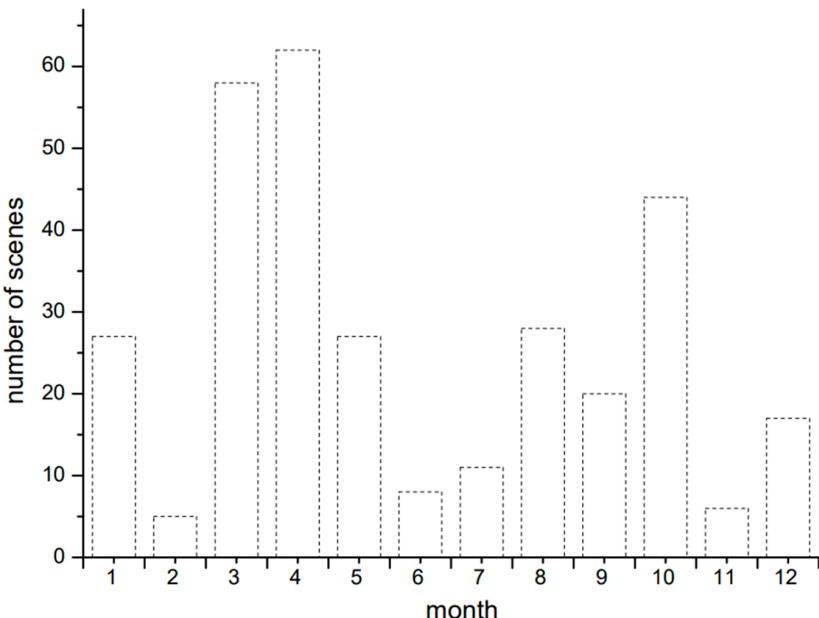

**Figure 6.** Statistics of 313 GF-2 scenes distributing in each month.

(2) Landsat-8 OLI LSR spectral adjustment.

The differences between the spectral bandpasses of GF-2 and OLI were slight in the first three bands, but quite evident in the NIR band (Figure 7). The OLI spectral bandpasses were used as a reference to adjust the GF-2 bands before cross-comparison. The spectral adjustment algorithm was realized through Hyperion atmospheric correction spectral curves convoluted by GF-2 and OLI RSRs, as in the procedure of the HLS project [65]. In total, 90 scenes of Hyperion were selected over China to provide the surface reflectance via the 6SV [66]. The atmospheric correction coefficients for each CMG pixel were retrieved using the auxiliary data (i.e., AOT, water vapor, and ozone content from the MODIS/Terra Surface Reflectance product (MOD09CMG), aerosol type from the MODIS/Terra Aerosol Optical Thickness product (MOD09CMA), and elevation from the Climate Modeling Grid Digital Elevation Model (CMGDEM)) at 0.05° and bilinearly resampled at 30 m for each Hyperion pixel. It was practical to reduce the size of retrieval surface reflectance pixels based on the principal components analysis (PCA) and unsupervised classification [65]. To further reduce the spectral artifacts, a smoothing step was performed before PCA transformation and the whole procedure was described as follows:

- The smoothing Hyperion surface reflectance image was obtained by applying the FLAASH polishing algorithm [67].
- A PCA was performed on the smoothing surface reflectance data in the spectral subset of less-atmospheric-affecting regions. A 99% of the variance was considered for the PCA transformation.
- An unsupervised ISODATA classifier was performed on each PCA-transformed scene and the centroid spectra of each class were extracted.

A total of 9604 centroid spectra were retrieved from the 90 scenes. The similar processes of the PCA and ISDATA methods were performed on the 9604 spectra to extract the final 338 spectra, representing the typical surface reflectance over the variety of ground types. A least-squares linear regression was used between equivalent spectral bands from GF-2 (explanatory variable) and OLI (dependent variable) reflectance to calculate the regression coefficients (Table 6). The GF-2 products were compared with the OLI reflectance products by applying these coefficients.

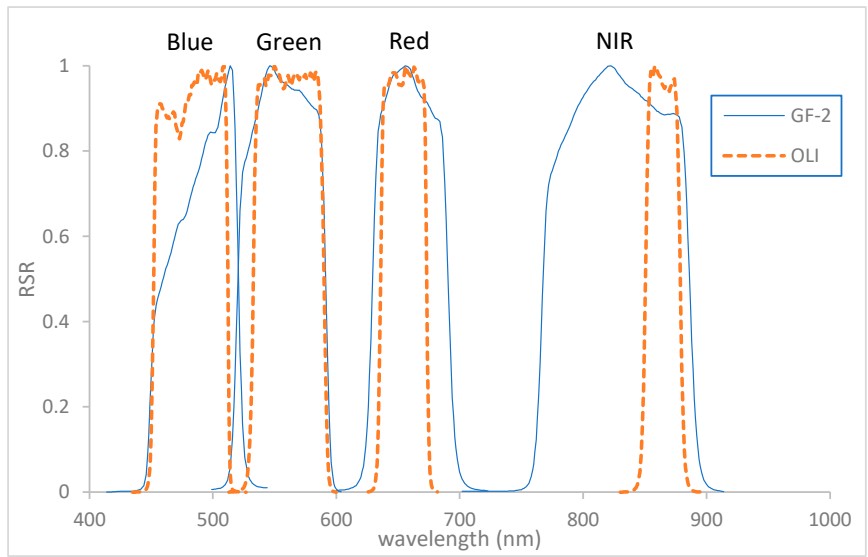

**Figure 7.** Relative spectral responses (RSRs) of GF-2 and Landsat-8 OLI sensors for similar bands.

**Table 6.** Coefficients of the linear regression bandpass adjustment (OLI = slope * GF2+ offset) and the mean residual.

| Band Name | Slope | Offset | $R^2$ | Residual |
|-----------|-------|--------|-------|----------|
| Blue | 0.99513 | −0.00326 | 0.99995 | 0.00120 |
| Green | 0.99922 | 0.00133 | 0.99998 | 0.00087 |
| Red | 1.00032 | −0.00171 | 0.99994 | 0.00139 |
| NIR | 1.01916 | 0.00657 | 0.99777 | 0.00752 |

To minimize the registration error between GF-2 and OLI, the image registration process was performed by using the Scale Invariant Feature Transform (SIFT) method, extracting SIFT features from OLI images. Further, both the GF-2 and OLI scenes were aggregated into 300 m to avoid residual registration error before the reflectance cross-comparison. A total of over 800,000 samples were selected to perform the cross comparison. The commonly used statistical metrics (i.e., accuracy (A), precision (P), and uncertainty (U)) were adopted to assess the spectral-adjusted GF-2 SR compared with OLI SR. The metrics are defined as follows [46]:

$$A = \frac{1}{n} \times \sum_{i=1}^{n} \varepsilon_i \tag{12}$$

$$P = \sqrt{\frac{1}{n-1} \times \sum_{i=1}^{n} (\varepsilon_i - A)^2} \tag{13}$$

$$U = \sqrt{\frac{1}{n} \times \sum_{i=1}^{n} \varepsilon_i{}^2} \tag{14}$$

where $n$ is the number of valid samples used for the comparison and $\varepsilon_i$ is the GF-2 SR minus OLI SR. The valid SR samples were compared, and the results are shown in Figure 8.

The overall performance of GF-2 SR showed good quality. The average values of A (accuracy) and U (uncertainty) for the whole SR range were 0.0027, 0.0019, 0.0011, and −000109, and 0.0011, 0.0010, 0.0017, and 0.0032, respectively. The values of A were relatively low and most of the U (uncertainty) values remained under the $S_{Lnd}x_{GF-2}$ line, which was defined as the quadratic sum of the specification of Landsat SR and that of GF-2. As the OLI SR specification was validated by AERONET measurements to 0.05ρ + 0.005 [46], the GF-2 SR specification could be set to 0.1ρ + 0.01. However, we noticed the evident discrepancy in the low SR values (<0.11) of the GF-2 NIR band. A similar phenomenon was also found in

both the SR products of the Landsat-5 TM and Landsat-7 ETM+ NIR bands when compared to MODIS SR [46]. Another discrepancy occurred in the high SR values (>0.32) of the GF-2 red band. This result supported the quality of GF-2 SR for most cases, except for the lower SR in the NIR band and the higher SR in the red band. This means that GF-2 SR was suitable for the qualitative studies for the land surface with medium- to high-density vegetation coverage but needed special concerns when applying to the cases of surfaces with very low and high reflectance (e.g., bright buildings, concrete surfaces, and water).

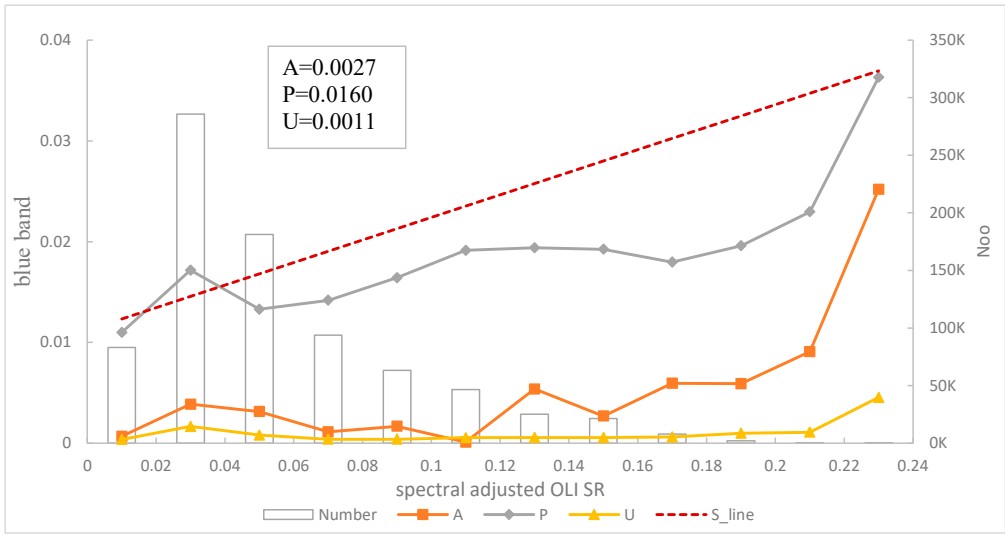

(**a**)

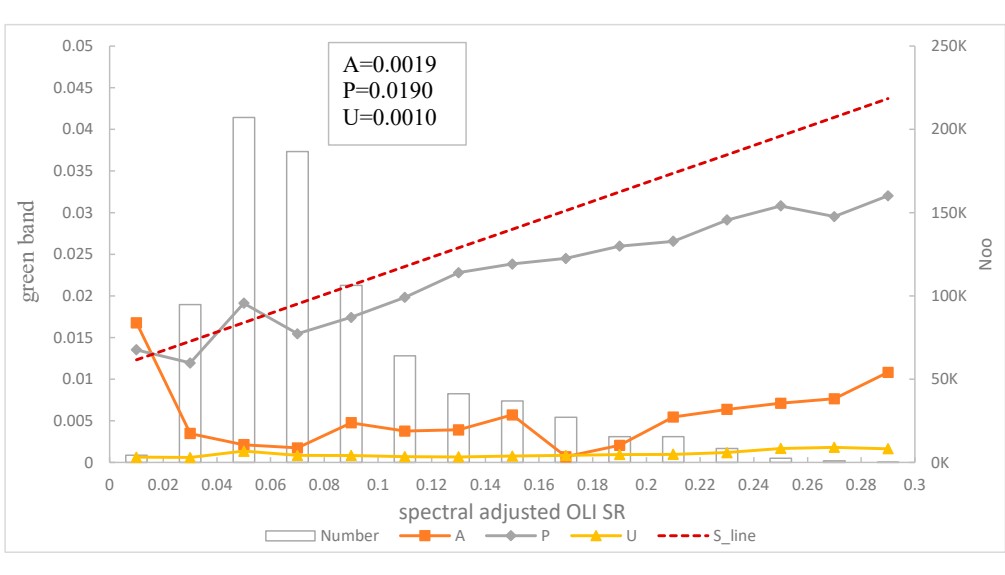

(**b**)

**Figure 8.** *Cont.*

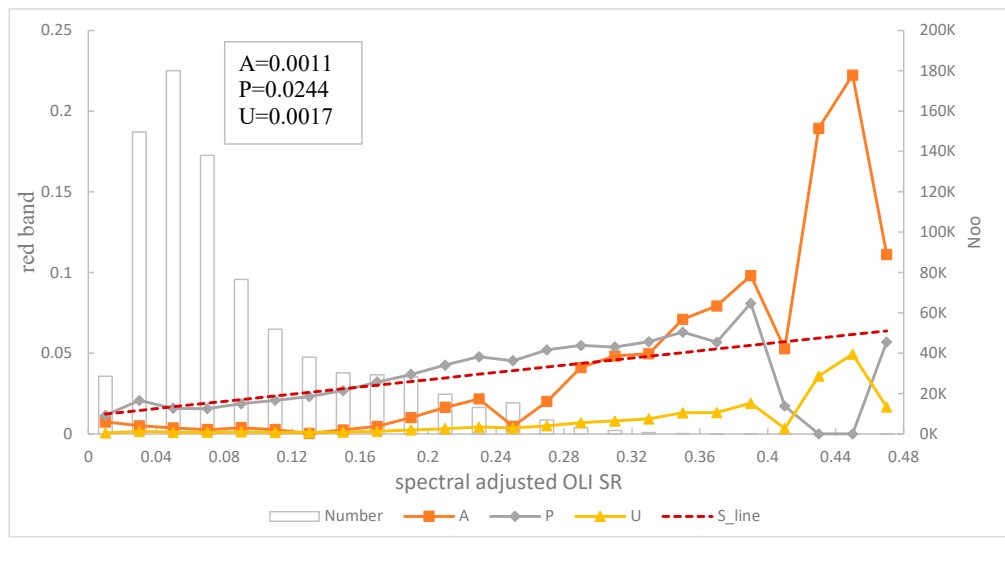

(**c**)

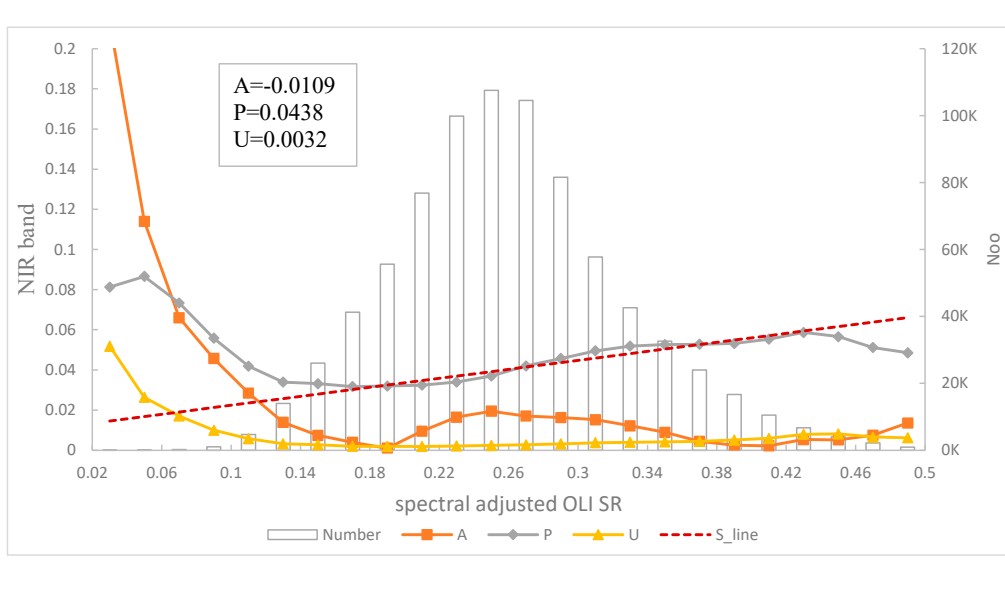

(**d**)

**Figure 8.** Cross-comparison SR APU graphs. Reported A (accuracy), P (precision), and U (uncertainty) of surface reflectance (SR) were computed from the cross-comparison of GF-2 versus the Landsat-8 OLI, used as reference (*x*-axis), showing in subgraphs for the blue (**a**), green (**b**), red (**c**), and NIR (**d**) bands, respectively. A, P, and U were computed per 0.02 bins and overall values (entire range) are written on the top-left. The specification of the cross-comparison ($S_{Lnd \times GF-2} = 0.112\rho + 0.0112$) is displayed in dashed line ("S_line"). SR reference values histograms are displayed in gray (right axis) with the corresponding number of occurrence (Noo), i.e., the number of considered pixels in each bin.

## 6.2. Processing Efficiency

Besides the LSR product evaluation, we also tested the system processing efficiency by inputting a large volume of satellite data for a large area. Three datasets were processed, as requested by other researchers, corresponding to 784, 593, and 592 scenes obtained in Henan province; Beijing, Tianjin, and Hebei province; and Hainan province, respectively. We processed these images on a single workstation equipped with a 3.6-GHz Intel Core i7-7700 processor and 8-GB RAM. As a comparison, the FLAASH and Sen2Cor was also employed to process the Landsat-8 OLI scenes and Sentinel-2B scenes, respectively, under the same processing environment. The processing details are listed in Table 7. It should

be noted that the volume of each standard scene was about 394 MB, 2.0 GB, and 800 MB for GF-2, OLI, and Sentinel-2B, respectively. The ACFrC showed a much faster processing speed (i.e., ~1.52 min per 100 MB) than the other two processors did (i.e., ~2.96 min and 8.55 per 100 MB by FLAASH and Sen2Cor, respectively). Regarding the inferior capacity of the workstation, all these software packages could perform much better by using a current mainstream configuration.

**Table 7.** Processing details for different data sources.

| Data Source | Regions | Processing Time Cost (h) |
| --- | --- | --- |
| GF-2 PMS, 784 scenes from 2016 | Henan province | ~84 (by ACFrC) |
| GF-1 PMS, 138 scenes from 2013, 2014, and 2017; GF-2 PMS, 316 scenes in 2017; and ZY-3 MUX, 139 scenes in 2013, 2014, and 2017 | Beijing, Tianjin, and Hebei province | ~60 (by ACFrC) |
| GF-2 PMS, 592 scenes from 2017 to 2021 | Hainan province | ~60 (by ACFrC) |
| Landsat-8 OLI, 10 scenes in 2021 | Beijing, Nei Mongol | ~10.33 (by FLAASH) |
| Sentinel-2B, 10 scenes in 2021 | Beijing, Shandong province | ~11.15 (by Sen2Cor) |

## 7. Discussion

Among the various relevant factors, four should be prioritized for the sustained development of ACFrC: the LUT accuracy, aerosol model selection, AC algorithms, and entire data processing chain optimization.

### 7.1. LUT Accuarcy

The accuracy of LUTs, which form the basis for AOT retrieval and AC, is determined by the inherent accuracy of the RT code, i.e., 6SV, itself as well as the LUT step design. The former has been extensively validated and is reported to have a very high value of approximately 0.4–0.6%, compared to standard benchmarks and other RT codes [66]. Thus, RT uncertainty has little impact on LUT accuracy and can be ignored. LUT design involves two aspects: gaseous transmission and atmospheric scattering. LUT accuracy certainly increases more or less with a simple narrowing of the interval between the input steps for 6SV; however, there is a considerable increase in time cost for LUT generation and searching, and the extremely large volume of LUTs themselves. We designed an LUT smart structure by establishing a tradeoff between accuracy and time cost. The accuracy was validated for GF-2 [68] as an example; the main findings are summarized as follows.

(1) The empirical formulas had relatively high accuracy, being eligible for gaseous transmission calculation. A relative maximum error of ~1.5% was obtained for the case of water vapor transmission in the NIR band because of its spectral range covering the water absorption wavelengths. The relative maximum errors for ozone and other gases (except for ozone and water vapor) were 0.1% and 0.09%, respectively. Figure 9 compares the gaseous transmissions calculated using the empirical formulas and by 6SV for the band having the largest difference.

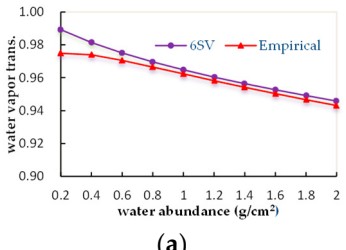 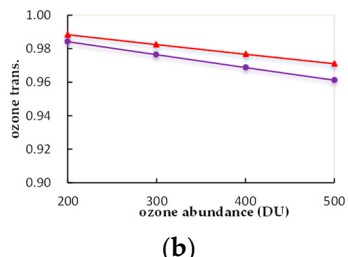 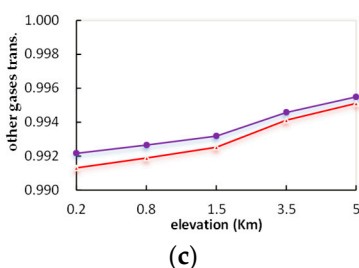

(**a**) (**b**) (**c**)

**Figure 9.** Comparisons of gaseous transmissions calculated using empirical formulas and by 6SV under various atmospheric conditions (water vapor transmission in infrared band (**a**); ozone transmission in blue band (**b**); and transmission of other gases in infrared band (**c**)).

(2)   The LSRs based on the LUTs had a high accuracy compared to those based on the direct inputting of the same atmospheric parameter to 6SV. As a case study, the LSRs computed based on the LUTs had absolute errors from 0.8% for the blue band to 0.5% for the NIR band when applied to GF-2 data. The LSRs based on the LUTs also exhibited superior accuracy compared to those retrieved using FLAASH (Table 8).

**Table 8.** Mean absolute errors for AC results provided by two AC methods compared with results given by 6SV directly.

| Band | FLAASH | LUTs |
|------|--------|------|
| Blue | $2.6 \pm 0.5\%$ | $0.8 \pm 0.010\%$ |
| Green | $1.8 \pm 0.4\%$ | $0.6 \pm 0.006\%$ |
| Red | $1.7 \pm 0.2\%$ | $0.5 \pm 0.003\%$ |
| NIR | $1.9 \pm 0.1\%$ | $0.5 \pm 0.002\%$ |

*7.2. Choice of Aerosol Model*

The aerosol model, which has been investigated by many researchers over many years, is crucial for AOT retrieval and AC from remote sensing data. The high-accuracy description of the aerosol properties of a certain location is challenging because of the complex characteristics of aerosols; this problem arises even when ground-based measurements such as AERONET observations are employed. However, the scientific community (e.g., the MODIS groups) has derived reliable AOT and LSR products from remote sensing data based on aerosol models proposed according to extensively distributed worldwide AERONET observations [21,22,46,69]. The aerosol model strategy varies among different AC software and LSR product systems. For LSR derivation for Landsats 4 to 7, a fixed "continental" aerosol type defined in 6SV was assumed by LEDAPS. As an improvement, a more flexible "urban clean" dynamic aerosol type was used for operational OLI LSR retrieval [46]. Even for MODIS, the MOD04 and MOD09 products have been retrieved based on slightly different aerosol models by two research groups [22,70], as mentioned in Section 3.1. Most commercial software (e.g., FLAASH, ATCOR, and ACORN) uses internally defined aerosol models depending on their RT codes [30,32,33]. Results of an AC inter-comparison exercise (ACIX) have shown that some AC processors can derive AOT with a higher accuracy but a poorer surface reflectance accuracy, whereas the opposite is true for other AC processors [51]. Thus, selection of the optimal aerosol model strategy is difficult, but this choice can be refined through product comparison. Given the operational and automatic running modes, for ACFrC, we adopted the four prescribed aerosol models that vary with time and worldwide location used in XBAER [23]. This choice was made because of the limited number of bands available for attempted aerosol model estimation in most of the multi-spectral remote sensing data. The preliminary LSR result products derived from GF-2 data also indicated that a reasonable accuracy could be achieved by AC based on these aerosol models. In future, comparative work will be conducted by using

different aerosol type sources from the MOD09 product, MOD04 product, and "urban clean" used in OLI SR retrievals.

### 7.3. AC Algorithm

A variety of AC algorithms have been developed and validated by previous studies [51]. These different algorithms have their specific advantages, and no single algorithm could achieve a high accuracy for all ground types. VNIR and the MODIS-based algorithms have been applied to HJ-1 CCD, showing a similar accuracy by comparison with AERONET-based retrievals over China [71]. However, AC algorithms should be compared via variety of ground types and a huge amount of data. Therefore, the use of a combination of AC algorithms for LSR retrieval is a potential avenue for future research. In addition, other issues also require consideration, such as AC over-bright pixels, topographic correction, and bidirectional reflectance distribution function (BRDF) correction.

The DB method was originally proposed for the retrieval of aerosol properties over brighter surfaces such as arid, semiarid, and urban regions [24,72]. The DB method for MODIS has been updated to newer versions and DB aerosol retrieval has been incorporated into Collection 6 products [25,73]. It is possible to extend the DB method to LSR retrieval from multi-spectral data. In this case, the critical step is the estimation of the LSR in the blue or deep-blue band through the LSR time series derived from other satellite data such as MODIS. Previous investigators have applied the DB method to HJ-CCD and OLI and derived satisfactory results [74,75]. Thus, the DB method for AC over bright areas is recommended in the future.

The topographic effect is more evident in remote sensing data with high spatial resolution than those with low or medium spatial resolution. The non-Lambertian reflectance behavior of many surfaces further increases the complexity of ground reflectance retrieval over mountainous regions. The practical strategy is to separate the topographic correction and BRDF normalization in sequential order [76]. Various topographic correction algorithms have been proposed, such as the cosine method and C-correction method applied in the Lambertian case and the modified Minnaert method involving the non-Lambertian correction [77,78]. These methods are easily applicable to operational data processing as their necessary parameters can be derived in a straightforward manner. However, the results derived by these methods still focus on certain study regions, and comprehensive verification for more regions is still necessary [79]. A prospective topographic correction method implemented in ATCOR computes the diffuse irradiance, direct irradiance, path radiance, and reflected terrain radiation from the neighborhood surface using the RT code [76]. However, for the application of this method, the spatial resolution of the DEM must match that of the image, and it is challenging to obtain a more detailed global DEM with a high spatial resolution of less than 10 m. In addition, BRDF correction is a more challenging task involving one scene only acquired at a fixed angle. Using a recently proposed operational method called "BREFCOR", it is possible to normalize the LSR derived from data acquired by wide-field-of-view sensors in a fixed imaging geometry [80]. Therefore, we plan to continue to verify the possible operational methods for topographic correction and BRDF correction and to implement these methods in ACFrC.

### 7.4. Data Processing Chain

The present version of ACFrC still uses a serial processing strategy for the data processing, i.e., pixel by pixel or subset by subset. In fact, many processing modules can be fully or partially implemented using a parallel strategy, e.g., the TOA reflectance calculation, masking, and AC modules. If only the data processing is spatially independent, parallel processing can greatly improve the data processing speed. A time cost in the range of a few seconds could be expected for one conventional remotely sensed scene (e.g., TM, ETM+, or GF-1/-2) when the upcoming version of ACFrC is completed. Along with parallel processing techniques, the distributed processing technologies that are extensively applicable to cloud computing will be combined to realize an online reflectance product

processing service from a variety of remote sensing data. This aim can be achieved through CPU-based parallelism driven by open multi-processing (OpenMP) technology and a GPU-based acceleration computation strategy on a single computational node. Within the whole processing chain, the message passing interface (MPI) executes communication among multiple nodes to realize the parallel processing of multiple images.

## 8. Conclusions

We proposed an operational reflectance product system named ACFrC for application to multi-source medium–high-resolution remote sensing data. Most multi-spectral satellites lack operational LSR products; however, our proposed system is likely to satisfy this demand. An improved AC algorithm, of which the original version was LaSRC for Landsat-8 OLI, was implemented in the present version of ACFrC, which was applicable to most multi-spectral remote data.

During the development of the proposed system, smart LUTs with an adequate accuracy based on 6SV code were established according to the specifications of the relevant sensors (e.g., their swaths, bands, and spectral response functions); these LUTs could be applied to the imaging of geometric and atmospheric scenarios. The LUTs could also be easily extended to new sensors simply by including their spectral response functions into the 6SV code before implementation. ACFrC is coded in the C++ language and automatically generates standard LSR products in HDF or GeoTIFF format. The preliminary results for GF-2 showed that the SR accuracy could be expected to be $0.112 \times \text{LSR} + 0.0112$ by cross-comparison of OLI SR products over China and adjacent regions. Furthermore, a data processing test using an ordinary computer revealed that the present version of the system satisfied the operational aims. There is no doubt that more validation work should be performed to evaluate the AC accuracy and processing efficiency of ACFrC for more satellite data, especially the four bands data. Moreover, in a future version of the system, AC algorithms over bright regions will be implemented and the fusion of different algorithms in the processing chain will be considered. Complex processing steps such as topographic and BRDF correction will certainly be included if robust and operational algorithms are developed by the scientific community. ACFrC will be available to scientific community for free soon after relevant materials are prepared (e.g., user manual, product description, website maintenance, etc.).

**Author Contributions:** Conceptualization, H.Z. and B.Z.; software, H.Z. and Z.F.; formal analysis, B.Z., H.Z. and D.Y.; writing—original draft preparation and revision, H.Z. and D.Y.; data preparation and rectification processing, B.L.; AC processing and software comparison, S.Z. All authors have read and agreed to the published version of the manuscript.

**Funding:** This research was supported by the National Natural Science Foundation of China (Grant No. 41771397), and Hainan Provincial Department of Science and Technology (Grant No. ZDKJ2019006, CAS Strategic Priority Research Program (Grant No. XDA19010402), and Key project of Aerospace Information Research Institute, CAS (Grant No. E0Z202010F).

**Institutional Review Board Statement:** Not applicable.

**Data Availability Statement:** The data presented in this study are available on request from the corresponding author. The data are not publicly available currently because most of the dataset especially the high-resolution data (e.g., GF-1, GF-2, and ZY-3) are belong to the second funding project and are only shared within project participants before the project finished.

**Acknowledgments:** We would like to thank the anonymous reviewers to help improve the quality of this work.

**Conflicts of Interest:** The authors declare no conflict of interest.

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
