# Peer review of "An Operational Atmospheric Correction Framework for Multi-Source Medium-High-Resolution Remote Sensing Data of China"

_remotesensing, doi:10.3390/rs14215590_

Round 1

Reviewer 1 Report

This is a very well written manuscript. I only suggest minor changes to improve the manuscript for revisions.

a) in the abstract, you do not need to mention specific atmospheric correction algorithms, such as ATCOR, ACORN, FLAASH, etc. You can just have a summary sentence about multiple atmospheric correction algorithms were previously developed and used for processing various types of data sets. 

b) In the introduction, you have an extensive citations of previous atmospheric correction algorithms. If possible, you can consider to add a few older atmospheric correction papers (to your manuscript)  for processing Landsat5 or Landsat4 data using LOWTRAN5 and LOWTRAN6 codes. You didn't cite any papers by Gao et al. on hyperspectral atmospheric corrections. There is a review paper by Gao et al. in 2009 RSE special edition in honor of Dr. Alex Goetz. 

c) p. 3, before Section 2, the paragraph starting with 'Therefore'. This word can be removed. The resulting sentence can start with 'In this study'.

d) p.5 and p.6, you mentioned 'LaSRC' algorithm, which was implemented for processing MODIS, Landsat8 OLI, Sentinel 2 MSI, VIIRS data, etc. The different implementations of LaSRC for different satellite data worked fine most of the time. However, the results from the MODIS implementation were not as great as the authors claimed in their published papers. The MODIS land surface reflectance data products routinely contained many clouds. The thin cirrus clouds were incorrectly removed as aerosols that causes large errors in reflectance data products for bands centered near 1.6 and 2.2 micron. Recently, a researcher provided me sample Sentinel 2 MSI land surface reflectance data products for a scene contained thin to thick smokes, the Sen2Con algorithm (a special version of LaSRC algorithm) performed poorly in removing smoke effects from MSI data. The algorithm was very slow to do atmospheric corrections for MSI data because of assumptions of surface reflectances and varying aerosol models through an iterative minimization retrieving process. 

My understanding is that your algorithm is based on LaSRC. It should work most of the time (may be ~90% of the time). However, under large smoke and aerosol loading situations, your algorithm should also perform poorly. In such situations, it is better to simply mask out the retrieval results, or assign very poor quality for the involved pixels. Alternatively, you can consider to adapt the 'Haze Removal' method implemented in a recent version of ATCOR4. The Haze Removal module should work much faster than Sen2Con for visible bands and produce more reliable surface reflectance data product in the visible (not necessarily for NIR and SWIR bands).

p.9, Table 2, the 6S code used a plane parallel model atmosphere. It is not good to do simulations for solar zenith and view zenith angles greater than about 70 degrees. The angular grid spacings for SZ, AN & VZ are a bit coarse, and should be decreased if possible. Foe SZ, you didn't have 0 degree (nadir direction). Does the 6SV code have trouble for SZ = 0 calculations?

p. 20, you mentioned 'DB' method, which requires assumptions of surface spectral reflectances based on previously retrieved values. Such assumptions can be a big problem, e.d., after rain, the surface reflectances can be much smaller than usual, the assumptions of small surface reflectance values can result in huge aerosol optical depths. Overall, both LaSRC and DB methods assumed spectral reflectance values for deriving aerosol information. The derived aerosol information is also used in the retrieving of surface reflectances. There is a 'chicken or egg' issue, or the 'catch-22' problem. What is assumed and what is retrieved?

Reviewer 2 Report

The article is devoted to the urgent problem of ground surface reflectance reconstruction from satellite data of medium and high resolution. All the most well-known approaches for solving this problem are described in the manuscript. In terms of methodology, there is no novelty in the work – well-known and well-established approaches for atmospheric correction are used. Nevertheless, the results of the work are of great practical importance, since they expand the capabilities of the atmospheric correction algorithm used for more than 10 satellite systems. To prove the correctness of the results obtained by the authors’ program, a cross-calibration has been performed with the ground surface reflectance data from the Landsat-8 OLI. The results obtained confirm the capabilities of the proposed program for creating standard products.

At the same time, there are several points that are not described in the manuscript that, in my opinion, should be added, and several questions regarding the text of the manuscript:

1.     P. 4: “The most important step in imagery AC is to determine the water vapor distribution and aerosol property.” It would be better to modify the sentence: “The most important step in imagery AC is to determine the water vapor distribution and the aerosol property for the wavelength range under study”. For other wavelength ranges, the situation can be observed in which another atmospheric gas will have high absorption.

2.     P. 5, Fig. 1: Different colors in the figure are not explained. It would be better to add explanation what do the colors mean.

3.     Have you taken into the account the influence of polarization when creating the LUT tables? It would be better to add the answer into the manuscript.

4.     Have you taken into the account the influence of refraction when creating the LUT tables? It would be better to add the answer into the manuscript.

5.     Am I right to understand that you have made the atmospheric correction with independent pixel simplification?

6.     P. 8: “The parameters in Eq. (9) are computed from the…”. If I am not mistaken, it is a misprint (it should be in Eq. (8)).

7.     P. 9: Table 2. “AOT”. Do you mean the AOT at λ = 550 nm in this table or the AOT for the bands under study?

8.     You use cloud and water masks. And what about pixels near the water and clouds? There is an adjacency effect near clouds and water. If you use the atmospheric correction algorithm with independent pixel simplification, then you can have pixels with bad reflectance recovery near clouds and water. I think that it would be better to add flags or masks indicating such regions or describe this problem for future investigations.

9.     P. 13: “LSR products derived from GF-2 satellite data obtained over China (313 scenes in total) for the period of 2015 to 2018 were considered (Figure 4).” It would be better to add more information about the scenes. What types of surfaces were considered in these scenes? Were there any surfaces near water? Near cloud and cloudless? How well do the Landsat-7 OLI and GF-2 data agree for these areas?

10.  P. 14: “snow and water free pixels”. Does the algorithm developed in this work applicable for snow-covered surfaces?

11.  P. 19, Fig. 7: The figure is not adequately reproduced, which makes it impossible to evaluate the results shown in it.

On the whole, the manuscript can be published after minor revision.

Reviewer 3 Report

The manuscript presents a procedure to perform atmospheric correction on medium resolution images from Chinese satellites, such as GaoFen-2. The content presented is interesting for the volume of the journal, and the methodology presented is clear and supported by sufficient academic citations.

The introduction is well presented and written.

The manuscript does not follow the usual system of separation into sections such as methods, results and discussion; but the distribution system used is correct and contributes to make the content of the work more understandable.

The title of the manuscript is intended to be both the acronym of the proposed method, and the use of capital letters seems to indicate that; it would probably be better to write the sentence correctly and put the acronym at the end, or even think of an acronym that is not as complex as OPAMER.

The methodology presented is correct. There are some details that could be improved.

First, Figure 1 and 6 should be presented in the journal style, indicating each quadrant with the letters a, b, c and d and its explanation at the bottom of the figure. A legend should also be presented with an indication of the meaning of the colors presented.

In section 3.3.1, it is possible that the conversion of the document has presented some formulas with characters that have disappeared and in their place there are squares; it is convenient to review this aspect (equations 4, 6, 7, 8, 9). On the other hand, the academic references where the equations come from are not indicated; some are known in the satellite work; others are not; but in all of them there should be a reference of where they were first formulated or a general reference where their origin appears.

The map in Figure 4 should include an indication of north and a graphic scale. China's location on the planet is well known, but the formatting details should be incorporated correctly. The blue color indicated in the caption appears as purple in the figure; revise this aspect in the final version.

In figure 7 there has been a displacement of the quadrants of the figures and they appear superimposed.

Finally, it should be noted that the hourly processing details of the images appear to show a throughput of about 10 images per hour, i.e. about 6 minutes per image. One could discuss the performance by comparing with other existing procedures, such as applying FLAASH on a Landsat image or applying Sen2Cor on a Sentinel image and show the comparative performance.

The final summary section should be conclusions.

It is not indicated whether the application is restricted for use or available to the scientific community that intends to use it for atmospheric correction.

In summary, a good paper, very interesting, well written and just lacking some minor details to modify.
